# TeLLMe what you see: Using LLMs to Explain Neurons in Vision Models

## Abstract

As the role of machine learning models continues to expand across diverse fields, the demand for model interpretability grows. This is particularly crucial for deep learning models, which are often referred to as black boxes, due to their highly nonlinear nature. This paper proposes a novel method for generating and evaluating concise explanations for the behavior of specific neurons in trained vision models. Doing so signifies an important step towards better understanding the decision making in neural networks. Our technique draws inspiration from a recently published framework that utilized GPT-4 for interpretability of language models. Here, we extend and expand the method to vision models, offering interpretations based on both neuron activations and weights in the network. We illustrate our approach using an AlexNet model and ViT trained on ImageNet, generating clear, human-readable explanations. Our method outperforms the current state-of-the-art in both quantitative and qualitative assessments, while also demonstrating superior capacity in capturing polysemic neuron behavior. The findings hold promise for enhancing transparency, trust and understanding in the deployment of deep learning vision models across various domains. The relevant code can be found in our GitHub repository.

## 1 Introduction

With the increasing prevalence of complex machine learning models in various domains of life, there has been a rising demand for interpretability. Understanding why these models make the decisions they do is crucial, not just from a research perspective, but also for ethical and societal reasons. Deep learning models, especially, are often seen as black boxes due to their complex and highly nonlinear nature. This challenge is particularly evident in the context of vision models, where the relationship between inputs (image pixels) and outputs (class labels) is often not straightforward.

In this paper, we address this challenge by developing a method for generating and evaluating short explanations for the behavior of individual neurons in trained vision models. To the best of our knowledge, this is the first time a Large Language Model (LLM) has been used this way. Additionally, by leveraging the power of LLMs, our method does not require training a specialized model, and instead offers a method to assessing the quality of the explanations at scale.

Interpreting the computations performed by deep networks has been split up into three major sub-areas: visualization, probing and explaining single neurons.

**Visualization**   The most common visualization technique for interpretability, is finding the input image that maximally activates a specific neuron (Erhan et al., 2009; Yosinski et al., 2015; Olah et al., 2017). However, beyond that, visualization has been used to offer additional insights into feature importance (Sundararajan et al., 2017; Zhou et al., 2015), and better understanding the self-attention mechanism of transformers (Vig, 2019; Brașoveanu & Andonie, 2020).

**Probing**   Another popular technique is using a secondary (often much smaller) classifier to estimate the type of information encoded in the hidden state of the main model. To that end, the secondary model is trained to predict a class (i.e. Part-of-Speech, Named Entities, Semantic Roles, Polarity, etc.), given the hidden states of a network layer from the main model. The efficacy of the prediction indicates what information is encoded. This technique was first introduced by Alain &

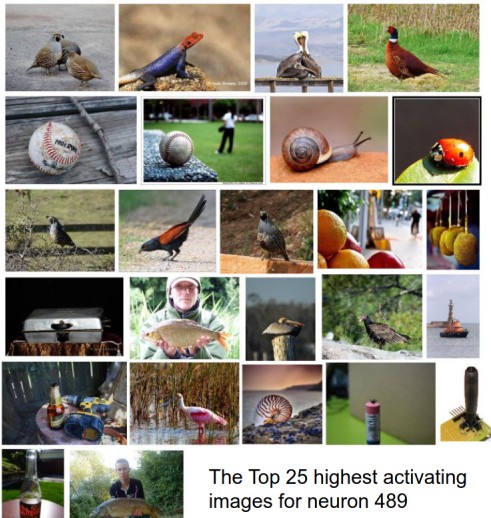

**Neuron Explanations**

**Clip-Dissect:** Mating

**GPT (Weight-Label)** **(ours):** the main thing this neuron does is find insects and other small creatures found in nature.

**GPT (Weight-CLIP)** **(ours):** the main thing this neuron does is find natural habitats and various types of animals found in those habitats.

**GPT (Caption-Activation)** **(ours):** the main thing this neuron does is find birds and insects. It also looks for bodies of water and sports equipment.

The Top 25 highest activating images for neuron 489

Figure 1: Left: The 25 highest activating images for an example neuron (neuron 489 of the second-last hidden layer) from an AlexNet model trained on ImageNet. Right: The generated explanation for this neuron, from the current SoTA method (Clip-Dissect) and the three methods proposed in this paper (GPT (Weight-Label), GPT (Weight-CLIP), GPT (Caption-Activation))

Bengio (2017) and has since been used to identify the role of specific neurons (Bau et al., 2020; Suau et al., 2020) and the level of complexity at different layers in transformer networks (Tenney et al., 2019; Voita et al., 2019; Raganato & Tiedemann, 2018; Nostalgebraist).

**Explaining Single Neurons** Most relevant to this work is the line of research that aims to find short explanations for individual neurons. Techniques employed to do so include manual inspection (Miller & Neo), creating a custom dataset and training a specialized classifier (Bau et al., 2017), and comparing the embedding similarities of images activating a specific neuron and text embeddings using a trained Vision-Language Model (Oikarinen & Weng, 2023). Additionally, Hernandez et al. (2022) trained a RNN to generate natural language descriptions of neurons in vision models, and Bills et al. (2023) used GPT-4 to generate short explanations of neurons in GPT-2 based on the words the neuron activated for. Notably, Bills et al. (2023) has also provided a way to **quantitatively** assess the quality of explanations. Namely, they use a secondary GPT-4 model to simulate the activations of a neuron given the explanation, and judge the quality of the explanations based on the correlation scores (here, and in the remainder of the paper we denote $100 *$ correlation coefficient as the correlation score) it achieved relative to the actual activations.

Our method is rooted in the framework introduced by Bills et al. (2023). However, we explain individual neurons in vision models, rather than language models, and do so via both the activations and the weights of the network. The key contributions of the paper are:

1. We generate short, easy-to-understand explanations of neuron selectivity in a trained vision model with two different techniques; firstly, using the weights of the model and secondly, using the image-caption / neuron activation pairs.
2. We are the first to propose a reliable & scalable explanation scoring algorithm for neurons in vision models.
3. We show that the proposed method works for other Vision models, including a ViT.
4. We show that our proposed methods are better in capturing polysemanticity, easier to understand, and quantitatively perform better than the current state-of-the-art.

The remainder of the paper is structured as follows: in Section 2 we highlight relevant related work and in Section 3, we present in detail our methods. We benchmark our methods against the current state-of-the-art in Section 4 and finally conclude in Section 5.

## 2 RELATED WORK

Out of the three main interpretability techniques described in Section 1, "Explaining Single Neurons" is most relevant to our work. As mentioned, interpretability can generally be sub-divided based on the techniques used to elucidate neuron explanations. Manually inspecting neurons and collecting specific datasets both introduce a large enough overhead for them to not be practical solutions for practitioners. Thus, we will be focusing on similarity based explanations and open-ended explanations.

### 2.1 SIMILARITY-BASED EXPLANATIONS

The recently published work by Oikarinen & Weng (2023) uses a trained Vision Language Model (VLM) to generate similarity based explanations in three steps.

1. First, the VLM is used to separately embed a set of probing images (which can be unlabeled) and a set of concept words. Subsequently, the concept-activation matrix is calculated by taking the inner product of each of the text and image embeddings.

2. Second, given a specific neuron that we want to explain, the activation of this neuron for each image in the probing set is recorded.

3. Lastly, the similarity between the activation vector from the second step and each column of the matrix in the first step is determined. The concept for the column with the highest similarity is used to describe the neuron.

This method has a good cost-quality trade-off. Whilst it is very cheap and fast to run, it has two shortcomings. Firstly, the one-term explanations it generates (i.e. "terrier", "feather", "nursery") are not very nuance and thus might miss subtleties of the neuron encoding. Secondly, there is a body of research documenting the prevalence of polysemantic neurons in deep learning models (Elhage et al., 2022; Olah et al., 2020). Naturally, it is hard to describe a neuron that looks for a multitude of potentially very different objects, with a single term.

### 2.2 OPEN ENDED EXPLANATIONS

The most important work in this area is the recently published paper "Language models can explain neurons in language models" by Bills et al. (2023). In their work, the authors used GPT-4 to generate human understandable explanations for individual neurons in GPT-2 (a large language model) based on how strongly they activate for various input words. Beyond simply generating explanations, they used a separate GPT-4 model to simulate the neuron activity for a test sentence based on the generate explanation; assessing the quality thereof by calculating the correlation between the real and the simulated activations. Though this is not the first work that is able to generate human understandable explanations for individual neurons, see Hernandez et al. (2022), to the best of our knowledge, it is the only work that does not require training a specialized model, generates high quality text, and most importantly, offers a method of assessing the quality of the explanations at scale.

## 3 METHOD

Our paper proposes two separate methods for explaining singular neurons in a trained vision model. First, we will describe the activations based method, and subsequently the weight based method. Both methods consist of an explanation generation component and a explanation assessment component.

Unless specified otherwise, we will use an AlexNet classifier trained on ImageNet and *gpt-3.5-turbo-0613* for all experiments. However, it is worth pointing out that given the results from Olah et al. (2020); Chughtai et al. (2023), and our experiments on Vision Transformers, our method is very likely to generalize to other vision models.

## 3.1 ACTIVATION-BASED EXPLANATIONS

Out of the two proposed methods, generating explanations based on neuron activations, is most similar to the work proposed by Bills et al. (2023). Since GPT-2 is a text based method, it is relatively easy to extract token-activation pairs of a specific neuron, and based on that to find and test explanations. However, since the GPT family of models does not yet support multi-modal inputs, and open source Vision-Language Models (VLMs) are not as capable as the GPT models, we need to use another model as the intermediary. To be more specific, we find captions for all images in the ImageNet (Russakovsky et al., 2014) validation set using BLIP (Li et al., 2022), and track the activation intensity specific neurons have for each image. Based on these Caption-Activation pairs we can generate and assess explanations. A diagram explaining the method can be seen in Figure 2 (the corresponding pseudocode for Figure 2 can be found in Appendix A.1).

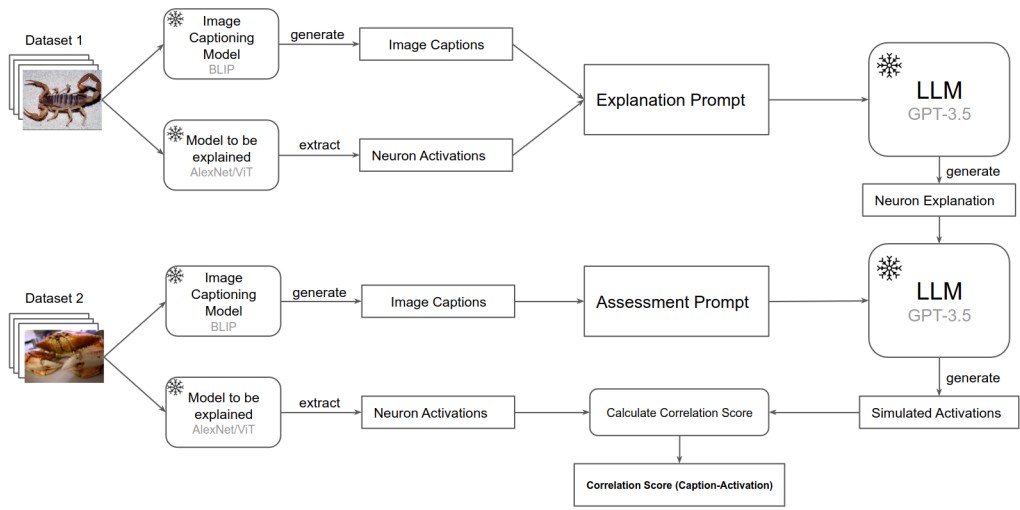

Figure 2: Overview of the proposed Caption-Activation method. An LLM is used to generate a human-readable explanation of an individual neuron's behavior, given a set of captions and activations. Then, an independent second LLM is used to generate the neuron's simulated (predicted) activations, given the generated explanation and a set of captions. The correlation between the simulated and actual activations are then calculated. Snowflake symbols indicate that models are frozen.

### 3.1.1 GENERATING EXPLANATIONS

To generate open-ended natural language explanations for the activities of a specific neuron, we utilize the well-documented ability of GPT to be an effective few-shot predictor. By showing it a number of few-shot examples, which entail a number Caption-Activation pairs for a neuron that isn't the neuron to be labeled, and then task GPT, based on the Caption-Activation pairs of the neuron that is to be labeled, to generate a short explanation. Details for determining the optimal number of few-shot examples and the size of the subset of Caption-Activation pairs shown to the model can be found in Appendix B.1.2.

Following Bills et al. (2023), we simplify the task at hand for GPT by expressing the activations as integers in the range $[0, 10]$. The full prompt used can be found in Appendix D.1.

### 3.1.2 ASSESSING EXPLANATIONS

Accurately determining how fitting these explanations are is more important than generating explanations for specific neurons. To do so, we prompt a GPT-3.5 model using image captions and its corresponding explanation to determine how strongly the neuron will activate for each caption. To simplify the task for GPT, we provide some few-shot examples, scale all activations in the range $[0, 10]$ and round them to be integers, similar to Bills et al. (2023). It is worth pointing out that we are using *gpt-3.5-turbo-0613* and thus are limited to a 4K context window. This means that at

any one time, we are only able to show GPT a subset of the available Caption-Activation pairs. We conduct some hyperparameter tuning in Appendix B.1.1 to determine the optimal number of few-shot examples and the size of the Caption-Activation subset. The full prompt for assessing neuron explanations can be found in Appendix D.2.

Finally, using the simulated activations, we determine the correlation to the true activations and use this as a measurement of success for the explanation. Since we will be introducing correlation scores with other targets than the neuron activations later on in this paper, from now on, we will refer to this one as Correlation Score (Caption-Activation).

One big challenge with using a subset of the images and few-shot examples is the variance in the correlation scores achieved by different runs. To fight this, we iteratively re-assess the same explanations until the 90% Confidence Interval (CI) of the average is $< \pm 5$ (see Algorithm 1).

---

**Algorithm 1** Explanation Scoring

---

**Require:** Neuron explanation.
    Scores = []
    **while** len(Scores) < 3 **or** CI(Scores) $\leq$ 5 **do**
        Scores append Score(Simulate(Neuron explanation))
    **end while**

---

where $CI(x) = T_{0.95, \text{sample\_size}-1} \cdot \frac{\text{sample\_std}}{\sqrt{\text{sample\_size}}}$ (T being the t-score corresponding to the critical value for a two-tailed test). Score(x) determines the correlation of the simulated activations and the actual activations, and Simulate(x), simulates the neuron activations based on the neuron explanation using GPT.

The main purpose of the algorithm is to save cost, since the alternative would be to run it for a constant (high) number of iterations and then take the average. Using the algorithm, we can stop as soon as it is sensible to do so (i.e. once the 90% CI of the average is $< \pm 5$.

### 3.2 WEIGHT-BASED EXPLANATIONS

Besides proposing an activation based explanation for neurons, we introduce a separate technique that utilizes the weights of the trained network. As will become evident, the last hidden layer, and all other layers have to be treated differently, thus we will first explain how the assessment and generation work for the last hidden layer, and subsequently, generalize this to the whole network.

Since we do not have access to the input-activation pairs when only using the weights to generate and assess explanations, we will extract the Weight-Label pairs (where the Labels are the ImageNet classes). This has the advantage of not relying on an image captioning model and not requiring us to pass a number of images through the network. However, just because a specific neuron is connected to a specific output label with a high weight, does not necessarily mean that the object related to the output class does indeed trigger high activations of the neuron. Furthermore, as we go deeper into the net (counting from the output layer), the features present will likely become increasingly abstract. Thus, it is not clear that the model, solely based on the class names and weights, is able to generate meaningful explanations for specific neurons. Figure 3 shows a diagram of the method (the corresponding pseudocode for Figure 3 can be found in Appendix A.2)..

#### 3.2.1 GENERATING EXPLANATIONS

The key difference to the technique used in Section 3.1.1, is that we use Weight-Label pairs, rather than Caption-Activations pairs. Other than that, we follow the same preprocessing as in Section 3.2.2, and the hyperparameters determined in Appendix B.2.2. The final prompt used can be found in Appendix D.4.

#### 3.2.2 ASSESSING EXPLANATIONS

Since the explanations are generated based on weights, we will assess them based on how well a secondary GPT model can predict the weights associated with each output class given a neuron

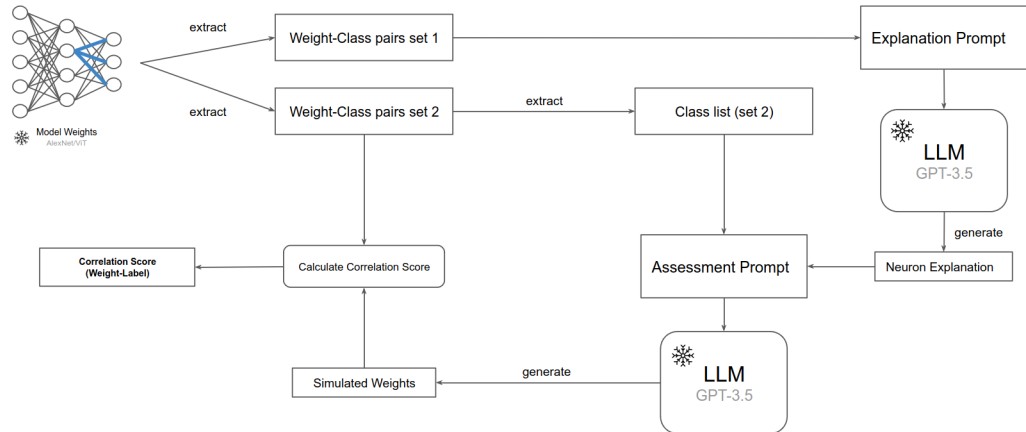

Figure 3: Overview of the proposed Weight-Label method. An LLM is used to generate a human-readable explanation of an individual neuron's behavior, given a set of labels and the magnitude of the weights that connect the neuron to be explained to that label. Then, a second, independent, LLM is used to simulate the neurons weights for another set of labels. The correlation between the simulated and actual weights are then calculated. Snowflake symbols indicate that models are frozen.

explanation. To simplify this task, we will set all negative weights to zero for now, scale the positive ones in the range $[0, 10]$, and convert them to integers.

Besides the above mentioned differences, we still utilize some few-shot examples, which each have a subset of the 1000 Weight-Label pairs (the subsetting is required, since we are using a 4K context window). The fine-tuning for these hyperparameters can be found in Appendix B.2.1, and the final prompt used can be found in Appendix D.5.

Following our set-up in Section 3.1.2, we quantify the explanation quality using the correlation score between the actual and predicted weights. Again, to make the results reliable, the same explanation is re-assessed, until the 90% CI is $< \pm 5$ (using Algorithm 1).

### 3.2.3 GENERALIZING TO OTHER LAYERS

As already mentioned, the last hidden layer is somewhat of a special case, as it is directly connected to the output layer, and with that, the class labels. Thus it is very easy to extract the Weight-Label pairs. However, since this can't be done for the remaining layers of the network, we propose three different techniques. Namely:

**Naively combining weights** The easiest method to implement is to estimate the weight connecting the, for example, second last hidden layer (or any other layer) to the output layer by taking the dot-product of the weight matrices.

$$W_{est} = W_t \cdot W_{t-1}...W_{target} \tag{1}$$

where $W_{est}$ is the estimated weight matrix, $W_t$ is the weight matrix connecting the $t - 1$ hidden layer to the $t$ (final) hidden layer.

Though this estimation is not perfect, as it disregards the activation function, and will make it harder for the model to determine the level of abstraction of the current neuron, it is a fast and cheap estimate that will serve well as a baseline. For the remainder of the paper we will refer to this method as Weight-Label.

**Using CLIP-Dissect labels as targets** Alternatively, it is possible to use simpler (and more importantly cheaper) methods to label all neurons of a specific layer with a simplistic explanations, and then use these explanation as target-weight pairs. In our experiments, we will be using CLIP-Dissect (Oikarinen & Weng, 2023), for this. This method will make it easier for GPT to determine the level

of abstraction required for the explanation, but might introduce inaccuracies for both explanation generation and assessment, as it relies on a secondary model. Furthermore, as we will show later, these simplistic methods do not capture polysemanticity well. For the remainder of the paper, we will refer to this method as Weight-CLIP.

**Using GPT-explanations as targets** Lastly, it is possible to label a whole layer using our method, before moving on to the next layer. The next layer can simply use the Weight-Explanation pairs to generate the next explanations. Though this method seems most promising, it has the obvious short-coming of being by far most expensive. To label the full last hidden layer in AlexNet would cost: # of Neurons $*$ # API calls per Neuron $*$ avg. prompt length $*$ API cost per 1K tokens $= 4096 *$ $25 * 2.5 * 0.0015 = 389$(USD). Due to the high cost, we did not test this method in this work but offer the method as a proposal.

## 4 EXPERIMENTS

To test the quality of the proposed methods, we will first generate explanations for neurons in an AlexNet model trained on ImageNet, via a plethora of different methods. These explanations will be assessed both quantitatively (based on the weight correlation and activation correlation) and qualitatively (by visualizing the subset of images in the ImageNet validation set that trigger the highest activation in the neuron we are explaining). Subsequently, we will show the quantitative results of our method when explaining neurons in a Vision Transformer. It is important to highlight that we report all results generated without cherry-picking.

Below we benchmark our techniques on Neuron 489 of the second last hidden layer. This neuron was randomly selected, and examples of other neurons can be found in Appendix C. Since the GPT-based methods required some few-shot examples, we hand-label 3 other neurons from the layer in question.

The techniques we will be benchmarking are:

- **CLIP-Dissect:** As our baseline, we use the CLIP-Dissect method proposed by Oikarinen & Weng (2023), using their official code base [1].

- **GPT (Caption-Activation):** Lastly, we use the BLIP generated image Captions and neuron activation magnitude pairs as features, as described in Section 3.1. The Correlation Score (Caption-Activation) simulates the activations of a neuron given the explanation.

- **GPT (Weight-Label):** As described in Section 3.2.3, this version of our method uses the Weight-output class pairs as features for generating the explanations and, the Correlation Score (Weight-Label), uses them for the simulation and subsequent assessment. For layers beyond the first hidden layer, the weight matrix from the current layer to the output layer is estimated via equation 1.

- **GPT (Weight-CLIP):** As described in Section 3.2.3, this version of our method uses the Weight CLIP-Dissect pairs as features. To that end, we used the CLIP-Dissect method to label all neurons in the network, and then simply extracted the weight CLIP-Dissect pairs, similarly to how the Weight-Label pairs are extracted from the last hidden layer. The Correlation Score (Weight-CLIP) aims to predict the Weight-CLIP pairs given a short neuron explanation.

Additional experiments using other LLMs than GPT-3.5 can be found in Appendix C.2.

### 4.1 QUANTITATIVE ANALYSIS

Table 1 shows the average Correlation Scores achieved by the various methods. Each method was used to generate 10 different explanations, and the highest scoring one is reported. The reason for doing so is that when actually using the methods to label a network, one has access to this information and presumably tries to find the best fitting explanation (rather than the average). As

---

[1]https://github.com/Trustworthy-ML-Lab/CLIP-dissect

| Method | Correlation Score | | |
|---|---|---|---|
| | **Caption-Activation** | **Weight-Label** | **Weight-CLIP** |
| CLIP-Dissect | 0.0% | 3.28% | 1.43% |
| GPT (Caption-Activation) | 15.43% | 31.20% | 6.51% |
| GPT (Weight-Label) | **16.11%** | **51.58%** | 8.27% |
| GPT (Weight-CLIP) | 5.47% | 9.38% | **22.05%** |

Table 1: The correlation scores achieved by various methods as assessed with three different targets each.

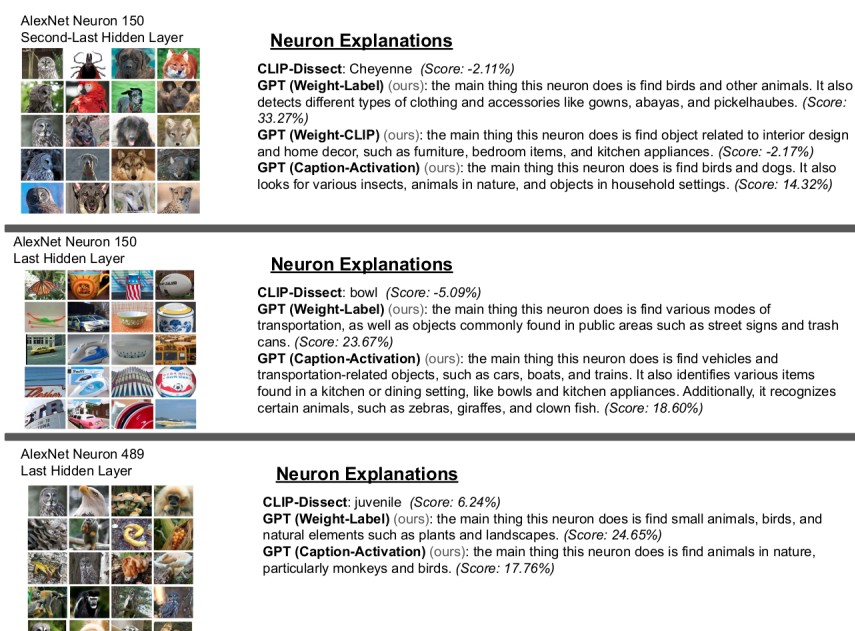

Figure 4: 24 of the highest activating images for three neurons from an alexnet model, side-by-side with the generated explanations, and the Correlation Scores (Weight-Label) these explanations achieved.

expected, the methods that use a specific feature pair tend to do best when assessed on that feature pair i.e. GPT (Weight-CLIP) does better than the other methods on the Correlation Score (Weight-CLIP). Overall, however, *GPT (Weight-Label)* clearly does best, with the best performance in two out of three metrics, and a decent performance on the last one. This is a somewhat surprising result as the Weight-Label pairs offer by no means definitive insight into which aspect of the image the neuron is activating for (i.e. a neuron might very well be activating for images containing planes, but only because these images tend to have the sky as the background). However, given the results from the table, it is clear that these implicit descriptions are good enough to generate a strong explanation.

It is worth pointing out that for the Correlation Score (Weight-Label) and (Weight-CLIP), we are only considering the positive weights (as mentioned in Section 3.2.2). All other weights have been zero-ed. The case with negative weights is analyzed in Appendix C.3.1.

## 4.2 Additional Qualitative Results

In addition to the qualitative results shown in Figure 11, we show 20 of the highest activating images for the three remaining neurons we explained in AlexNet, side-by-side, with the generated explanations as well as the Correlation Score (Weight-Label) (See Figure 4.

### 4.3 EXPLAINING A VIT

Following Oikarinen & Weng (2023), we conducted additional experiments of our method an the last hidden layer of a ViT-B-16 (trained on ImageNet). We used the Weight-Label approach since it worked best for our main experiments. To provide some additional insights, we also report the Accuracy (Acc), Mean Squared Error (MSE) and Mean Absolute Error (MAE), in addition to the achieved correlation scores (Corr) (all of which are calculated for the simulated weights vs the true weights). The neurons were selected randomly, and we report all results without any cherry picking.

| Neuron | Scores | | | |
|---|---|---|---|---|
| | **Corr.** | **Acc.** | **MSE** | **MAE** |
| 50 | 18.66% | 8% | 13.34 | 3.22 |
| 75 | 19.10% | 30% | 12.98 | 2.66 |
| 122 | 50.47% | 28% | 5.96 | 1.84 |
| 150 | 23.11% | 34% | 11.18 | 2.34 |
| 457 | 26.78% | 24% | 7.20 | 2.12 |
| 489 | 22.08% | 44% | 6.62 | 1.70 |
| 746 | 30.89% | 38% | 5.04 | 1.52 |
| **Avg.** | **27.30%** | **29.43%** | **8.90** | **2.2** |

Table 2: The performance of our method on a number of randomly-selected neurons from the last hidden layer of a Vision Transformer.

As can be seen in Table 2, the proposed method does a good join explaining a number of neurons from the last hidden layer of a vision transformer. This is a good indication that the method will generalize to other models and architectures.

## 5 CONCLUSION

Our experiments show that our proposed method of explaining the behavior of specific neurons in vision models performs well both quantitatively and qualitatively. This is demonstrated by the higher correlation scores in comparison to current state-of-the-art methods and the qualitative analysis of the explanations. To the best of our knowledge, this is the first time that a Large Language Model has been used to create human readable explanations of specific neurons in a vision model.

While we focused on AlexNet and ViT in this work, this methodology could be extended to other vision models. A larger language model could potentially provide more nuanced and accurate explanations, as indicated by the experiments in Bills et al. (2023).

However, the proposed methods are not without limitations. The biggest drawbacks are that some explanations are vague, using the GPT-3.5 API is costly (especially when labeling a complete network) and contrary to CLIP-Dissect, the method requires some hand-labeled examples (though, as we have shown in our experiments, only 2 per layer).

In conclusion, we have presented a method for explaining the behavior of specific neurons in vision models using GPT-3.5. Our method is intuitive, scalable, and can be extended to various vision models. While we acknowledge its limitations, we believe it represents a significant step forward in the area of machine learning interpretability, particularly in understanding complex deep learning models. By making these models more interpretable, we can foster greater trust and transparency, which is crucial as these models become increasingly prevalent in various domains of life.

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

# A  ARCHITECTURE

## A.1  PSEUDOCODE FOR THE ACTIVATION-CAPTION METHOD

---
**Algorithm 2** Pseudocode for the Activation-Caption method

---
**Require:** Two sets of images ($X^1$ & $X^2$).
**Require:** An Image Captioning model ($\Omega_{\text{captioning}}$)
**Require:** The model to be explained ($\Omega_{\text{main}}$)
**Require:** A Large Language Model (LLM)
  Extract the neuron activations of the model to be explained ($\Omega_{\text{main}}$) from both image sets ($h_{\Omega_{\text{main}}(X^1)}$ & $h_{\Omega_{\text{main}}(X^1)}$; denoted as $H^1$ & $H^2$ respectively)
  Scale the activations to be integers in range 0 to 10.
  Use the Image Captioning model ($\Omega_{\text{captioning}}$) to generate captions for the images in $X^1$ and $X^2$ (denoted as $C^1$ and $C^2$ respectively).
  Create the Explanation Prompt by using the caption ($C^1$) activation ($H^1_{\text{neuron\_idx}}$) pairs. (denoted as ExpPrompt)
  Generate the neuron explanations $E = \mathcal{D}_{\text{test}} LLM \,(\text{ExpPrompt})$
  Scores = []
  **while** len(Scores) < 3 **or** CI(Scores) ≤ 5 **do**
    Create the Assessment Prompt using the captions ($C^2$). (denoted as AssPrompt)
    Simulate the neuron activations $\tilde{H}^2_{\text{neuron\_idx}}$ using LLM (AssPrompt)
    determine the correlation score between $H^2_{\text{neuron\_idx}}$ and $\tilde{H}^2_{\text{neuron\_idx}}$
    append the correlation score to Score
  **end while**
  The final Score for the explanation is the average of Scores.

---

The Pseudocode above corresponds to the diagram in Figure 2.

## A.2  PSEUDOCODE FOR THE WEIGHT-LABEL METHOD

---
**Algorithm 3** Pseudocode for the Activation-Caption method

---
**Require:** The Weights of a trained Deep Neural Network ($W$)
**Require:** The class names of the dataset for which the DNN was Trained ($Y$)
**Require:** A Large Language Model (LLM)
  Extract the Weight-Label pairs from the set of weights $W$ and the class names $Y$ with one of the methods described in Section 3.2.3
  Scale the weights to be integers in range 0 to 10.
  Split the Weight-Label pair list into two subsets, denoted as $\text{WL}^1$ and $\text{WL}^2$.
  Use $\text{WL}^1$ to create the Explanation Prompt (denoted as ExpPrompt).
  Generate the neuron explanations $E = \mathcal{D}_{\text{test}} LLM \,(\text{ExpPrompt})$
  Scores = []
  **while** len(Scores) < 3 **or** CI(Scores) ≤ 5 **do**
    Create the Assessment Prompt using the class names from ($\text{WL}^2$). (denoted as AssPrompt)
    Simulate the neuron weights corresponding to the class names in the AssPrompt using the LLM.
    determine the correlation score between the generated weights and the actual weights (taken from $\text{WL}^2$).
    append the correlation score to Score
  **end while**
  The final Score for the explanation is the average of Scores.

---

The Pseudocode above corresponds to the diagram in Figure 3.

# B HYPERPARAMETER TUNING

## B.1 HYPERPARAMETER TUNING FOR ACTIVATION BASED EXPLANATIONS

### B.1.1 ASSESSING EXPLANATIONS

The hyperparameters we will tune are the "number of few-shot examples" $\in [1, 2, 3]$ and the "number of Caption-Activation pairs" $\in [10, 25, 50, 75]$. It is worth noting that since we are limited to a 4K context window and the descriptions are longer than the class names, we are not able to include as many examples pairs as we can with the weight based method.

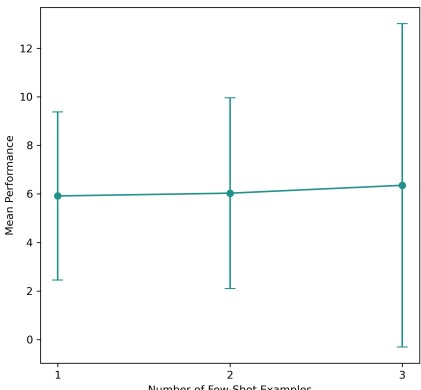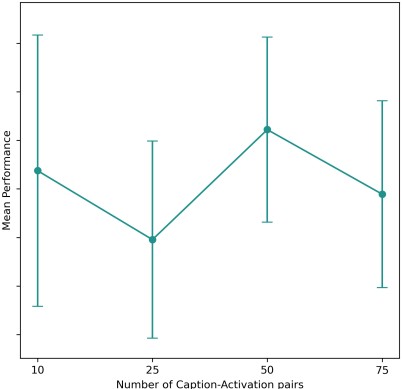

Figure 5: The average Correlation Score (Activation) for each hyperparameter. Error bars are the 90% Confidence Intervals

All unique combinations of the hyperparameters are run 25 times each. Figure 5 shows the average and 90% Confidence Interval of the achieved correlations scores. The average and CI are determined by concatenating the scores achieved by all runs with, for example, 2 Few-Shot examples and the different combinations of the number of Caption-Activation pairs. This means that the graph will not necessarily show the single best performing option, but offer some insight into which options perform best most consistently. It is worth pointing out that all runs where the prompt exceeded the 4K context window were discarded, which is why the error bars grow when the values get larger.

Overall, it seems that using 3 few-shot examples and 50 Caption-Activation pairs works best most robustly. However, this combination often times results in too long a prompt, thus we will be using 2 few-shot examples and 50 caption-activation pairs.

Additionally, we test if the way we present and pick the Caption-Activation pairs for the few-shot examples has an impact upon the performance. It is worth highlighting that this is only done for the few-shot examples, as it would leak information when used for the actual target neuron. Here, we will be comparing 5 different strategies:

- **Baseline** As a baseline, we simply randomly select the Caption-Activation subset.
- **Random 50/50** The first advanced strategy we explore is randomly selecting 50% Caption-Activation pairs where the activation is non-zero and 50% Caption-Activation pairs where the activation is zero.
- **Highest 50/50** Similarly to the option above, we randomly select 50% Caption-Activation pairs where the activation is zero, and then fill the remaining 50% with the highest activating Caption-Activation pairs. The logic for doing so is to more clearly show the model the range of activations ($[0, 10]$), it is supposed to simulate.
- **Highest 75/25** Following the same strategy as above, here we fill 75% of the subset with the highest activating Caption-Activation pairs, and the remaining 25% with zero activating Caption-Activation pairs.

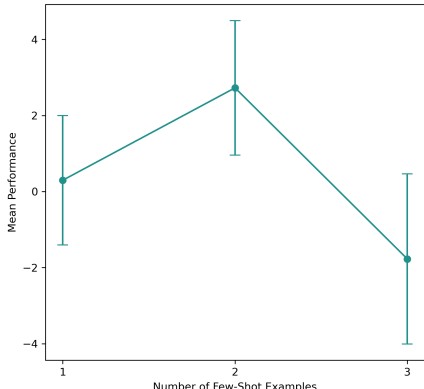 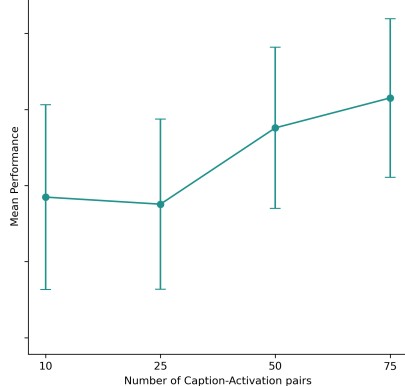

Figure 6: The average Correlation Score (Activation) for each of Explanation Generation hyperparameters. Error bars show the 90% CI.

- **Highest 90/10** Lastly, we further increase the ratio of the above method to 90% highest activating and 10% zero activating.

| Method | Correlation Score (Activation) |
|---|---|
| Baseline | $3.45 \pm 3.44$ |
| Random 50/50 | $3.44 \pm 3.93$ |
| Highest 50/50 | $4.21 \pm 3.27$ |
| Highest 75/25 | $0.81 \pm 3.41$ |
| Highest 90/10 | $-0.53 \pm 4.52$ |

Table 3: The average and 90% CI for each of the methods over 50 runs.

As can be seen in Table 3, the **Highest 50/50** has both the highest average performance and the smallest confidence interval. Thus, going forward, we will be using this strategy for creating the few-shot prompts when simulating neuron activity in the Caption-Activation set-up.

### B.1.2 GENERATING EXPLANATIONS

The two hyperparameters we will tune in the assessing explanation prompt are the "number of few-shot examples" $\in [1, 2, 3]$ and the "number of Caption-Activation pairs" $\in [10, 25, 50, 75]$. Again, the 4K context window is the limiting factor for both of these hyperparameters.

For each of the hyperparameter combinations 5 unique explanations were generated, and each of the explanations assessed using Algorithm 1 until the 90% Confidence Interval was $< \pm 5$. Subsequently, for each of the features, the scores achieved by all combinations with the other feature were concatenated, and the average and 90% Confidence Interval is shown in Figure 6. Similarly to the section above, we will not pick the single best performant combination, but rather the most robustly well performing one. In this case, that is 2 few-shot examples and 75 caption-activation pairs. Again, the error bars tend to grow with as the value increase, because we disregard all runs where the prompt exceeded the 4K context window.

Before tuning the hyperparameters, it is worth pointing out that, similarly to Bills et al. (2023), we first show the model all caption-activation pairs, and then all non-zero caption-activation pairs. Additionally, since this made hand labeling the few-shot examples much easier, we sorted these non-zero activation pairs in ascending order.

Again, we will experiment presenting the Caption-Activation pairs for the few-shot examples, and in this case, also for the target neuron, in 3 different strategies:

- **Baseline (Random)** Our baseline strategy is to randomly select Caption-Activation pairs.

- **Random (highlight non-zero)** Following Bills et al. (2023), for this strategy, we will randomly select Caption-Activation pairs, and below the list of pairs, show the model all non-zero Caption-Activation pairs.

- **Top** Lastly, we will test only showing the model the highest activating Caption-Activation pairs.

| Method | Correlation Score (Activation) |
|---|---|
| Baseline (Random) | $4.28 \pm 4.06$ |
| Random (highlight non-zero) | $2.78 \pm 3.09$ |
| Top | $7.88 \pm 2.66$ |

Table 4: The average and 90% CI for each of the methods for 5 explanations each. The explanations are scored until the 90% CI is $< \pm 5$ (See Algorithm 1

As can be seen in Table 4, the **Top** strategy performed best. Thus, going forward, unless specified otherwise, this is the strategy we will be using to present Caption-Activation pairs when generating explanations.

### B.2 HYPERPARAMETER TUNING FOR WEIGHT BASED EXPLANATIONS

#### B.2.1 ASSESSING EXPLANATIONS

To determine the best choice of hyperparameters for assessing weight-based neuron explanations, we run all combinations of "the number of few-shot examples" $\in [1, 2, 3, 4, 5, 6, 7]$, "the number of few-shot Weight-Label pairs" $\in [25, 50, 75, 100, 150, 200, 250]$ and "the number of predicted classes" $\in [25, 50, 75, 100, 150, 200, 250]$ 50 times each. Subsequently, for each value in each feature, we concatenate the results and plot the average value, with the 90% Confidence Interval as error bars, in Figure 7. All input combinations where the prompt exceeded 4K tokens were disregarded, which is why the Confidence Interval for the values towards the right tend to be higher.

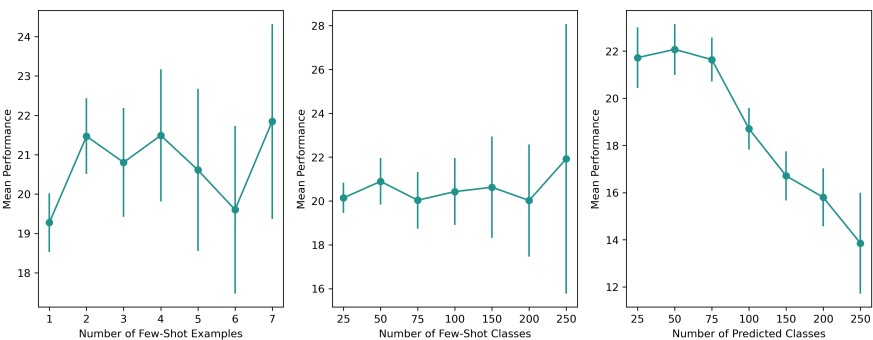

Figure 7: The average correlation scores of a number of combinations of the three hyperparameters for Assessing Explanations.

The combination with the best trade-off of prompt length and robust performance seems to be: 2 Few-Shot examples, with 50 Weight-Label pairs each, and predicting 50 Weight-Label pairs for the neuron currently being assessed. It is worth pointing out that this might not be the overall most performant option, but because of the size of the error bars, we are trying to use the most robustly well performing option. Unless specified otherwise, this is the combination we will use for all Weight based explanations throughout the paper.

The obvious next point of improvement is to select specific Weight-Label pairs to be shown in the few-shot examples. It is worth pointing out that the strategy explained below is not applied to the Weight-Label pairs that are to be predicted, as this would lead to information leakage.

Here, we examine the impact of assessing varying proportions of zero weight and non-zero weight classes to our few-shot examples. The intent is to mitigate potential performance degradation resulting from the sparsity of high positive weights, a condition exacerbated by restricting model prediction to 50 out of 1000 weights at a time. The five strategies tested are:

1. **Baseline** This strategy utilizes a simple random selection method for populating the Weight-Label pairs in the few-shot examples.

2. **Highest 50/50** This method populates the few-shot examples' Weight-Label pairs with a 50:50 ratio, comprising of the highest weights for the specific neuron and a randomly sampled assortment of zero weights.

3. **Random 50/50** Here we populate the Weight-Label pairs for the few-shot examples with 50% randomly sampled non-zero weights and 50% randomly samples zero weights.

4. **Random 75/25** Same as the previous, but with a 75/25 ratio.

5. **Random 90/10** Same as the previous, but with a 90/10 ratio.

Similarly to the experiments above, we run each of the options 50 times and report the average and the 90% Confidence Interval in Table 5. As can be seen in the table, populating the class-weight pairs with 75% randomly selected positive weights and 25% randomly selected zero-weights leads to the best average performance and a good Confidence Interval.

| Method | Mean Correlation Score |
|---|---|
| Baseline | $26.03\pm4.53$ |
| Highest 50/50 | $25.74\pm4.01$ |
| Random 50/50 | $30.21\pm4.52$ |
| Random 75/25 | $\mathbf{30.74}\pm4.19$ |
| Random 90/10 | $29.72\pm4.93$ |

Table 5: The average and 90% CI for each of the methods over 50 runs.

Finally, we investigate the correlation (if any) of the correlation scores achieved by the neurons used as few-shot examples, and the correlation score achieved by the neuron currently being assessed.

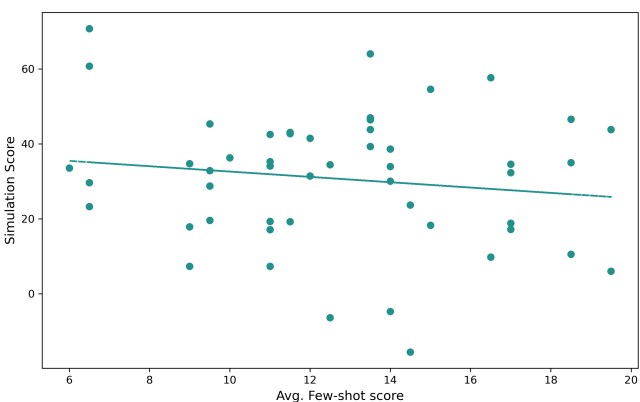

Figure 8: A scatter plot with a best fit line showing the relation between the avg. correlation score of the few-shot examples used, and the correlation score achieved by the neuron currently being assessed.

Figure 8 shows the scatter plot and corresponding best-fit line for a neuron. Interestingly enough, there is no positive correlation, but rather a slight negative one. However, numerically, the $0.14$ correlation has a p-value of $0.32$ and thus is not statistically significant.

Thus our strategy for assessing neurons is to use two few-shot examples with 50 Weight-Label pairs each (the class weight pairs consist of 75% positive weights and 25% zero weights, which are both randomly sampled), and predict 50 Weight-Label pairs, for the neuron currently being assessed, with each pass.

### B.2.2 GENERATING EXPLANATIONS

To find the best combination of hyperparameters for generating explanations, we run all combinations of: "the number of few-shot examples" $\in [1, 2, 3, 4, 5, 6]$ and "the number of Weight-Label pairs for the few-shot examples and actual prediction" $\in [25, 50, 100, 200]$, 3 times each using Algorithm 1. All prompts that exceed the context window length of 4K tokens are excluded (which is why the error bar tends to grow when using more few-shot examples or more Weight-Label pairs).

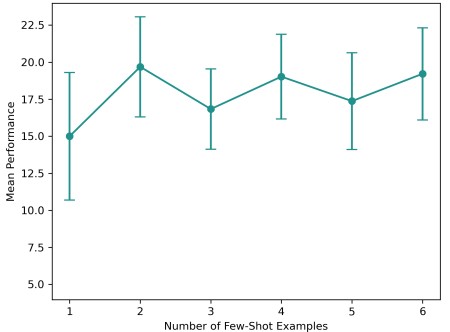 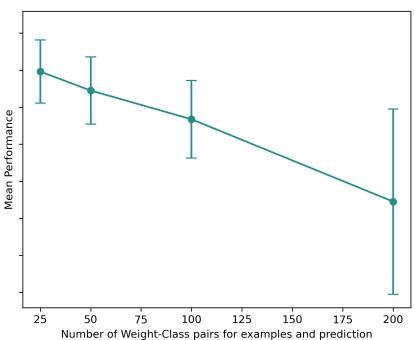

Figure 9: The average correlation scores of a number of combinations of the two hyperparameters for Generating Explanations. The error bars represent the 90% Confidence interval.

Interestingly enough, as can be seen in Figure 9, the average score achieved seems to be pretty independent of the number of few-shot examples used, whilst negatively correlating the number of Weight-Label pairs. This is likely because we show the model the top-n Weight-Label pairs, and thus, when the number grows too large, it will be harder to focus only on the important ones.Thus, going forward, unless specified otherwise, we will be using two few-shot examples with 25 weight-class pairs each for all explanation generated by the weight based method.

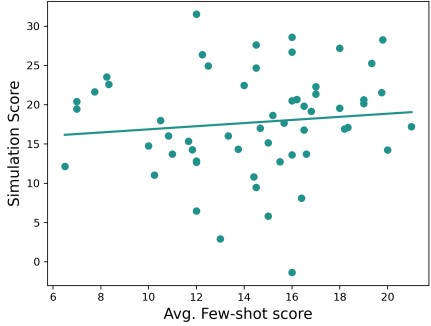

Figure 10: he scatter plot and best-fit-line for the avg. score of the few-shot examples (X) and the score achieved by the generated explanation (y).

Next, we will be testing if there is any correlation between the average score achieved by the few-shot examples and the score achieved by the neuron currently being explained. As can be seen Figure 10, this time there is indeed a positive correlation. Intuitively this makes sense, as the quality of the few-shot examples provided to the model should have an influence on the quality of the description generated. However, when testing numerically, the correlation of 0.11 only has a p-value of 0.42 and is thus not statistically significant.

## C    ADDITIONAL EXPERIMENTS

### C.1    QUALITATIVE ANALYSIS OF THE METHODS

Since generating the quantitative scores is part of our method, and thus not necessarily a perfectly objective measurement of success, we will also offer a qualitative analysis of the explanations in this section. To that end, we extracted the 25 images with the highest activation of Neuron 489 of the second last layer (the same neuron as above), from the ImageNet validation set. The images can be found in Figure 11.

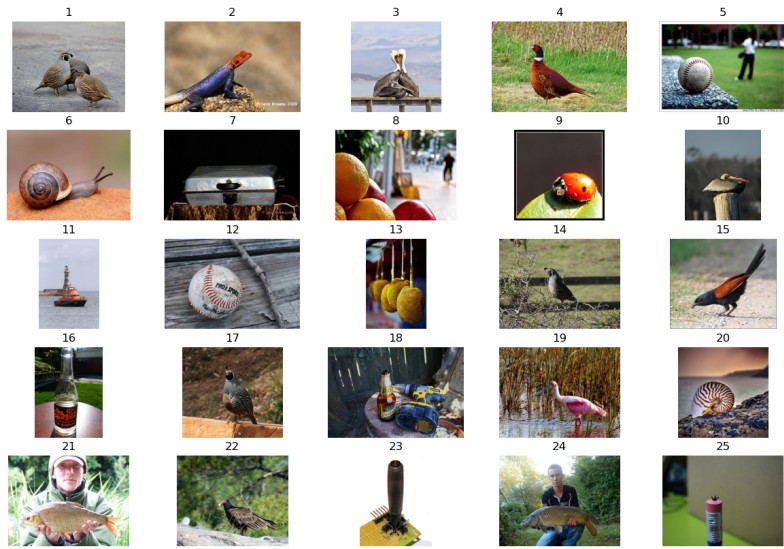

Figure 11: The 25 images of the ImageNet validation set that lead to the highest activation of Neuron 489 in the second last layer.

Before diving into the actual analysis, it is worth pointing out that the neuron clearly seems to be polysemantic, reacting to a plethora of objects that range from birds to fish and baseballs. Also, simply showing the images that lead to the highest activation does by all means not give a clear picture of the performance; it does, however, offer an additional data-point to Table 1.

The generated explanations corresponding to the result in Table 1 are:

- **CLIP-Dissect:** Mating (For the simulations, this is changed to: "the main thing this neuron does is find mating."
- **GPT (Weight-Label):** the main thing this neuron does is find insects and other small creatures found in nature.
- **GPT (Weight-CLIP):** the main thing this neuron does is find natural habitats and various types of animals found in those habitats.
- **GPT (Caption-Activation)** the main thing this neuron does is find birds and insects. It also looks for bodies of water and sports equipment.

There are a number of things worth pointing out. We will first comment on the explanations of the individual methods, and how well they fit the images in Figure 11, and subsequently compare them where relevant.

**CLIP-Dissect**    The explanation generated by the CLIP-Dissect method is very short (thus we augmented it when evaluating it via our simulation) and seems to fit two of the 25 images pretty well (Number 1 and 3), especially considering the low cost of the method. However, this highlights well the shortcoming of the method. Namely, it generates one word (or one term) explanations for neurons that often times are polysemantic, and thus the explanation might only capture a fraction of what the neuron is looking for.

**GPT (Weight-Label)**   With the impressive scores this explanation achieved in Table 1, it was expectable that the explanation is somewhat vague, as it has to cover a vast number of terms with a very short text. However, it does seem to capture most of the images in Figure 11 well, with the notable exceptions of the baseballs, boat, suitcase, bottles, food, eraser and electrical tool. Thus it does not seem to have fully captured the polysemanticity of the neuron. However, it is worth pointing out that the top 25 images are not necessarily representative of all, and the number of images explained is in line with the results of Table 1.

**GPT (Weight-CLIP)**   This method has a similar explanation to the one generated by *GPT (Weight-Label)*, but interestingly, performed considerably worse in Table 1. Nonetheless, it seems to fit about the same number of images as *GPT (Weight-Label)*, but, based on the description, seems to do so mostly because of the backgrounds.

**GPT (Caption-Activation)**   Before talking about the actual description, it is worth pointing out that, as described in Section 3.1.1, this explanation is based on the captions of the images in the ImageNet validation set that had the highest activation for the specific neuron; thus the explanation is certainly expected to fit the pictures best, as it is in-sample. As expected, the explanation fits almost all pictures exceedingly well, with the only exceptions being the suitcase, bottles, food, eraser and electrical tool.

Overall, the methods tend to cover varying numbers of concepts of the clearly polysemantic neuron.

## C.2   ALTERNATIVE LLMS

Recently, with the growing popularity of Large Language Models, an impressive open source online community has been trying to keep up, and even though the OpenAI models are still considered state-of-the-art, the recently release LLAMA-2 Touvron et al. (2023) family of models achieves strong results on popular benchmarks, and have the big advantage of being run locally. To test whether these models can be used as a cheaper alternative for labeling single neurons, we repeat the experiments from Section 4. Unfortunately, the model (we use a quantized LLAMA-2-70B) fails to keep the required structure when assessing the neurons (i.e. repeating each token and generating the relevant score). Since we used the same prompt as we did for our original experiments, there is a chance that adjusting the prompt can solve this issue. Furthermore, it is possible to restrict the output space of the model to the subset that is relevant, or to simulate one token at a time. However, we will leave these two approaches to future work.

It is also worth pointing out that even though Bills et al. (2023) clearly showed that GPT-4 performs better than GPT-3, we used GPT-3 in this paper because of the vastly cheaper cost.

## C.3   POSITIVE & NEGATIVE WEIGHTS

### C.3.1   NEGATIVE WEIGHTS

As mentioned both in Section 4 and Section 3.2.2, thus far we have restricted the *GPT (Weight-Label)* and *GPT (Weight-CLIP)* methods to only positive weights, zero-ing the negative ones. It is worth noting that this was not necessary for *GPT (Caption-Activation)* as AlexNet uses ReLU activation functions, and thus all values were positive anyway. In this section we will explore to what extent and how we can incorporate the negative weights. This is important since in neural networks the negative passes can be as influential as the positive ones by inhibiting the network to predict a certain class.

To first determine to what extent our method can be used to reliably explain the negative weights of a specific neuron, we will flip all weights, zero-ing the previously positive ones and use the *GPT (Weight-Label)* method to try to explain the negative aspect of the neuron.

Using the same set-up as in Section 4.1, we get the following explanation: **"the main thing this neuron does is find objects related to clothing and accessories, including formal wear, swimwear, and headwear. It also seems to have a focus on items found in specific places, such as libraries and toyshops. Additionally, it has a strong activation for certain dog breeds."**. To assess the quality of this explanation, we will determine the Correlation Score (Weight-Label) and list the 25

classes associated with the lowest weights. For negative weights, getting the neuron activations doesn't make much sense, as most of them are zero.

The explanation achieved a Correlation Score (Weight-Label) of $31.36\%$ which is smaller than that of the positive weights ($51.58\%$), but still relatively high.

To better highlight the fit of the explanation, we will list the 25 classes in two different columns. One for these that we deem to fit the explanation well, and one for the others.

| Good Fit | | Poor Fit | |
|---|---|---|---|
| Weight | Class | Weight | Class |
| -0.071 | German short-haired pointer | -0.064 | siamang |
| -0.063 | standard poodle | -0.055 | gorilla |
| -0.060 | cuirass | -0.054 | chiffonier |
| -0.059 | breastplate | -0.054 | fire screen |
| -0.059 | nack brace | -0.054 | gibbon |
| -0.059 | Newfoundland dog | -0.052 | entertainment center |
| -0.057 | bearskin | -0.052 | oboe |
| -0.055 | bathing trunks | -0.050 | steel drum |
| -0.054 | academic gown | | |
| -0.052 | hoopskirt | | |
| -0.051 | English foxhound | | |
| -0.051 | wig | | |
| -0.050 | suit of clothes | | |
| -0.050 | sarong | | |
| -0.050 | pajama | | |
| -0.049 | fur coat | | |

Table 6: The top 25 lowest weights and their associated classes for Neuron 489 of the second last hidden layer, hand-sorted into whether the explanation fits well or not.

Table 6 shows the 25 lowest Weight-Label pairs of Neuron 489 of the second last hidden layer. They were hand sorted into whether the explanation fits them well or not. Again, it is worth highlighting that solely looking at the top 25 classes (out of 1000) does not necessarily give a complete picture. The explanation seems to fit the majority of the classes well, though it notably could have been more specific in describing metal based / armor clothing items. However, since only very few classes out of the 1000 correspond to such a specific label, describing them as clothing items might simply be more economical. It is interesting to see that the three main groups of classes it seems to have not included in the explanation are primates (i.e. *siamang, gorilla, gibbon*), furniture (i.e. *chiffonier, fire screen, entertainment center*) and cylindrical metal objects (i.e. *oboe, steel drum*). Again, this clear multitude of distinct classes is evidence for the neurons polysemantic nature, and thus for the need for open-ended text explanations (rather than single-term explanations).

### C.3.2 POSITIVE & NEGATIVE WEIGHTS COMBINED

In our last experiment, we will combine the positive explanation generated by *GPT (Weight-Label)* and the negative explanation generated above by hand. Subsequently, we will scale the positive weights in range $[0, 10]$ and the negative weights in range $[-10, 0]$, rounding both to integers.

The final explanation used is: **"the main thing this neuron does is find insects and other small creatures found in nature.** It reacts negatively to **objects related to clothing and accessories, including formal wear, swimwear, and headwear. It also seems to have a** negative **focus on items found in specific places, such as libraries and toyshops. Additionally, it has a strong** negative **activation for certain dog breeds.".** We tried to keep the manual adjustments made when combining the two explanations to a minimum, and highlighted those adjustments in gray above.

We repeatedly simulated the positive and negative weights of each class given the explanation, with GPT until the 90% Confidence Interval was $< \pm 5$. This lead to a final correlation score of $49.63\%$. Considering that the positive prompt was able to explain the positive weights with a score of $51.58\%$ and the negative one was able to explain the negative weights with a score of $31.36\%$, and more

importantly, predicting both positive and negative weights at the same time seems intuitively much harder than doing so one-by-one, this score is certainly impressive. Again, we followed the same set-up as in Section 4, where we assess the model 10 times, and pick the best one.

As a closing thought on the experiments we want to highlight that that different types of Correlation Scores in Table 1, as well as Section C.3.1 and Section C.3.2 are not directly comparable, as they use different targets. For example, intuitively, it seems easier to summarize $1,000$ Weight-Label pairs than $10,000$ Caption-Activation pairs (this is the size of the subset we took from them ImageNet validation set). However, since the *GPT (Weight-Label)* explanation performed best for both the Caption-Activation and Weight-Label Correlation Scores, this is the method we recommend using.

## C.4 SECOND-LAST HIDDEN LAYER

### C.4.1 NEURON 150

| Method | Correlation Score | | |
|---|---|---|---|
| | **Weight-Label** | **Weight-CLIP** | **Caption-Activation** |
| CLIP-Dissect | -2.11% | 2.86% | -5.00% |
| GPT (Weight-Label) | 33.27% | -3.07% | 11.12% |
| GPT (Weight-CLIP) | -2.17% | 11.54% | -0.49% |
| GPT (Caption-Activation) | 14.32% | 0.34% | 13.59% |

Table 7: The correlation scores achieved by various methods as assessed with three different targets each.

- **CLIP-Dissect:** The main thing this neuron does is find cheyenne.
- **GPT (Weight-Label):** The main thing this neuron does is find birds and other animals. It also detects different types of clothing and accessories like gowns, abayas, and pickelhaubes.
- **GPT (Weight-CLIP):** The main thing this neuron does is find objects related to interior design and home decor, such as furniture, bedroom items, and kitchen appliances.
- **GPT (Caption-Activation):** The main thing this neuron does is find birds and dogs. It also looks for various insects, animals in nature, and objects in household settings.

## C.5 LAST HIDDEN LAYER

When finding explanations for the last hidden layer, we can simply extract the Weight-Class pairs from the network weights and thus don't rely on the speical techniques introduces in Section 3.2.3. This means that *GPT (Weight-Label)* and *GPT (Weight-CLIP)* as well as *Correlation Score (Weight-Label)* and *Correlation Score (Weight-CLIP)* are, respectively, identical. Thus, for the below sections, we will denote the techniques as *GPT (Weight-Label)* and *Correlation Score (Weight-Label)*.

## C.6 NEURON 489

| Method | Correlation Score | |
|---|---|---|
| | **Weight-Label** | **Caption-Activation** |
| CLIP-Dissect | 6.24% | 5.63% |
| GPT (Weight-Label) | 24.65% | 10.63% |
| GPT (Caption-Activation) | 17.76% | 38.04% |

Table 8: The correlation scores achieved by various methods as assessed with three different targets each.

- **CLIP-Dissect:** The main thing this neuron does is find juvenile.
- **GPT (Weight-Label):** The main thing this neuron does is find small animals, birds, and natural elements such as plants and landscapes.

- **GPT (Caption-Activation):** animals in nature, particularly monkeys and birds.

### C.6.1   NEURON 150

| Method | Correlation Score | |
| --- | --- | --- |
| | **Weight-Label** | **Caption-Activation** |
| CLIP-Dissect | -5.09% | 1.34% |
| GPT (Weight-Label) | 23.67% | 7.87% |
| GPT (Caption-Activation) | 18.60% | 13.42% |

Table 9: The correlation scores achieved by various methods as assessed with three different targets each.

- **CLIP-Dissect:** The main thing this neuron does is find bowl.
- **GPT (Weight-Label):** The main thing this neuron does is find various modes of transportation, as well as objects commonly found in public areas such as street signs and trash cans.
- **GPT (Caption-Activation):** vehicles and transportation-related objects, such as cars, boats, and trains. It also identifies various items found in a kitchen or dining setting, like bowls and kitchen appliances. Additionally, it recognizes certain animals, such as zebras, giraffes, and clown fish.

## D   PROMPTS

In this section we will present the prompts used for the three different methods. We tried to keep the prompts as similar as possible to the ones used in Bills et al. (2023), but had to slightly adjust them for the weigh and clip based methods.

Below, we will firs present the Caption-Activation prompts for explanation assessment (Section D.2 and explanation generation (Section D.1, then move on the presenting the Weight-Label prompts for explanation assessment (Section D.5) and explanation generation (Section D.4), and finally present the Weight-CLIP prompts for explanation assessment (Section D.5) and explanation generation (Section D.6).

## D.1 Prompts for Activation Based Explanations

Similarly to the prompt for assessing the explanations, the final prompt used for generating them consists of the system prompt (highlighted in olive), the few-shot questions (highlighted in blue), the few-shot replies (highlighted in purple) and the target neuron prompt (highlighted in teal). As pointed out in Appendix B.1.2, the Caption-Activation pairs for both the few-shot examples and the target neuron are subsampled by choosing the ones with the highest activation and ordering them in ascending order when presented to GPT.

---

We're studying neurons in a neural network. Each neuron looks for some particular thing in an image. Look at the descriptions of images the neuron activates for and summarize in a single sentence what the neuron is looking for. Don't list examples of words.
  The activation format is description<tab>activation. Activation values range from 0 to 10. A neuron finding what it's looking for is represented by a non-zero activation value. The higher the activation value, the stronger the match.

Neuron 2
Activations:
<start>
a cat walking through a field of leaves<tab>3
a basket of apples and oranges with a sign<tab>3
a yellow umbrella with a wooden pole and a yellow cloth<tab>3
[...]
a colorful wheel with a black rim<tab>6
a dog laying in the grass with a tennis ball<tab>9
two dogs sitting on a bench with the words usa written on it<tab>10
<end>
Explanation of neuron 2 behavior: the main thing this neuron does is find objects and animals on grass. It also looks for british food and beaches.

Neuron 1
Activations:
<start>
a dog running with a frisbee in its mouth<tab>5
a bug on the ground with sand in the background<tab>5
a tractor plow in the snow near a tree<tab>5
[...]
a spider with a red eye on its back<tab>9
a woman holding a snake on her shoulder<tab>9
a toy axamant in a tube in a tank<tab>10
<end>
Explanation of neuron 1 behavior: the main thing this neuron does is find trucks and food. It also finds insects.

Neuron 489
Activations:
<start> a piano keyboard with four keys and a case<tab>4
a golf ball on a green field with a man in the background<tab>4
a fish in a tank with gravel and water<tab>4
[...]
a bird standing on the ground in the grass<tab>8
a bunch of fruit hanging from a rack<tab>10
a snail crawling on a rock with a blurry background<tab>10
<end>
Explanation of neuron 489 behavior: the main thing this neuron does is find

---

where <tab> is the tab token, and "[...]" indicates that we truncated the prompt.

## D.2 PROMPTS FOR ACTIVATION BASED ASSESSMENT

The final prompt used is composed of the system prompt (highlighted in olive), the few-shot questions (highlighted in blue), the few-shot replies (highlighted in purple) and the target neuron prompt (highlighted in teal). As mentioned in Appendix B.1.1, the specific Caption-Activation pairs for the few-shot examples are selected with the **Highest 50/50** strategy (which means that out of the 50 Caption-Activation pairs used for each few-shot example, we randomly select 25 Caption-Activation pairs where the activation is zero, and the 25 Caption-Activation pairs with the highest activations.

---

We're studying neurons in a neural network. Each neuron looks for some particular thing in an image. Look at an explanation of what the neuron does, and try to predict how it will fire for each image description.
The activation format is description<tab>activation, activations go from 0 to 10, unk indicates an unknown activation. Most activations will be 0.

Neuron 1
Explanation of neuron 1 behavior: the main thing this neuron does is find trucks and food. It also finds insects.
Activations:
<start>
a man in a wet suit and snorg diving<tab>unk
a woman standing on a walkway with a boat in the background<tab>unk
a bird sitting on a log in the woods<tab>unk
[...]
two women with punk hair and piercings<tab>unk
a black dog with a long hair sitting on a step<tab>unk
<end>
<start> a man in a wet suit and snorg diving<tab>2
a woman standing on a walkway with a boat in the background<tab>0
a bird sitting on a log in the woods<tab>t0
[...]
two women with punk hair and piercings<tab>0
a black dog with a long hair sitting on a step<tab>0
<end>

Neuron 2
[...]

Neuron 489
Explanation of neuron 489 behavior: the main thing this neuron does is find insects, birds, mushrooms and unusual structures in natural settings.
Activations:
<start>
a bathroom sink with a mirror and a towel rack<tab>unk
a tool kit sitting on top of a table<tab>unk
a field with a hay bale and a tree<tab>unk
[...]
a group of women in black shirts and pink aprons<tab>unk
a laptop computer sitting on a table<tab>unk
a dog sitting in the grass with a leash<tab>unk
<end>

---

where <tab> is the tab token, and "[...]" indicates that we truncated the prompt.

### D.3 PROMPTS FOR WEIGHT-LABEL BASED EXPLANATION ASSESSMENT

To keep consistency, we will color-code the below prompt with the same colors as the prompts above, namely: the system prompt in olive, the few-shot questions in blue, the few-shot replies in purple and the target neuron prompt in teal.

The Weight-Label pairs for the few-shot examples are selected based on the **Random 75/25** strategy, where we randomly select 25% zero-weight pairs and 75% non-zero-weight pairs.

---

We're studying neurons in a computer vision neural network. Each neuron looks for some particular thing in an image, and based on that influences the prediction probability for the available classes. Based on a short explanation of the neuron, you should try to predict the weight associated with each class.The description need not be related to the objects themselves, but what might typically be found in the same image as the object (i.e. fish & water).We present the classes as class-description<tab>unk, where it will be your task to predict weight inplace of the unk tokens.A neuron finding what it's looking for is represented by a high positive value. The higher the value the stronger the match.

Neuron 0
Explanation of neuron 0: the main thing this neuron does is find cliffs, wild snakes (both sea and land) and african animals.
Activations:
<start>
cliff<tab>unk
grocery store<tab>unk
sturgeon<tab>unk
[...]
matchstick<tab>unk
hartebeest<tab>unk
volcano<tab>unk
<end>

<start>
cliff<tab>1
grocery store<tab>5
sturgeon<tab>2
[...]
matchstick<tab>2
hartebeest<tab>0
volcano<tab>5
<end>

Neuron 2
Explanation of neuron 2: the main thing this neuron does is find large buildings and bridges. It also looks for vehicles.
Activations:
<start> seawall<tab>unk
garbage truck<tab>unk
bassoon<tab>unk
[...]
academic gown<tab>unk
ocean liner<tab>unk
gorilla<tab>unk
<end>

<start>
seawall<tab>5
garbage truck<tab>5
bassoon<tab>0
[...]
academic gown<tab>2
ocean liner<tab>5
gorilla<tab>3
<end>

Neuron 489
Explanation of neuron 489: the main thing this neuron does is find insects and other small creatures found in nature.
Activations:
<start>
kite<tab>unk
jellyfish<tab>unk
screw<tab>unk
[...]
crash helmet<tab>unk
espresso<tab>unk
sawmill<tab>unk
<end>

---

where <tab> is the tab token, and "[...]" indicates that we truncated the prompt.

## D.4 Prompts for Weight-Label Based Explanation Generating

In the example prompt below, we color coded: the system prompt in olive, the few-shot questions in blue, the few-shot replies in purple and the target neuron prompt in teal.

We're studying neurons in a computer vision neural network. Each neuron looks for some particular thing in an image. Look at the classes that the neuron is positively contributing towards and summarize in a short text what the neuron is looking for (the key thing most of classes have in common). Don't list examples of classes. Keep in mind that these are images, so the information connecting them might not be related to the object itself, but rather to what can be found in images of the object (i.e. fish & water).

The activation format is token<tab>weight. Activation values range from 0 to 10. A neuron finding what it's looking for is represented by a high positive value. The higher the value the stronger the match. You should not list objects, but rather try to find what they have in common.

Neuron 0
Activations:
<start>
toucan<tab>6
water ouzel<tab>6
worm snake<tab>6
[...]
ostrich<tab>9
water snake<tab>9
cliff dwelling<tab>10
<end>

Explanation of Neuron 0 behavior: the main thing this neuron does is find cliffs, wild snakes (both sea and land) and african animals.

Neuron 1
Activations:
<start>
threshing machine<tab>6
transverse flute<tab>6
Border collie<tab>7
[...]
punching bag<tab>9
axolotl<tab>10
dishwasher<tab>10
<end>

Explanation of Neuron 1 behavior: the main thing this neuron does is find american farm related items.

Neuron 489
Activations:
<start>
quill pen<tab>7
ruler<tab>7
smoothing iron<tab>7
[...]
grasshopper<tab>9
snail<tab>9
lacewing fly<tab>10
<end>

Explanation of Neuron 489 behavior: the main thing this neuron does is find

where <tab> is the tab token, and "[...]" indicates that we truncated the prompt.

## D.5 PROMPTS FOR WEIGHT-CLIP BASED EXPLANATION ASSESSMENT

To keep consistency, we will color-code the below prompt with the same colors as the prompts above, namely: the system prompt in olive, the few-shot questions in blue, the few-shot replies in purple and the target neuron prompt in teal.

We're studying neurons in a computer vision neural network. Each neuron looks for some particular thing in an image, and based on that influences the prediction probability for the available classes. Based on a short explanation of the neuron, you should try to predict the weight associated with each class. The description need not be related to the objects themselves, but what might typically be found in the same image as the object (i.e. fish & water). We present the classes as class-description<tab>unk, where it will be your task to predict weight inplace of the unk tokens. A neuron finding what it's looking for is represented by a high positive value. The higher the value the stronger the match.

Neuron 0
Explanation of neuron 0: the main thing this neuron does is find agriculture related animals, grains and items. It also looks for wild rare animals and hunting dogs.
Activations:
<start>
elk<tab>unk
orioles<tab>unk
plaque<tab>unk
[...]
implements<tab>unk
dome<tab>unk
sculptures<tab>unk
<end>

<start>
elk<tab>0
orioles<tab>1
plaque<tab>2
[...]
implements<tab>1
dome<tab>0
sculptures<tab>1
<end>

Neuron 2
Explanation of neuron 2: the main thing this neuron does is find large dogs and cats.
Activations:
<start>
bottles<tab>unk
terrier<tab>unk
wildlife<tab>unk
[...]
monarch<tab>unk
terrier<tab>unk
pitcher<tab>unk
<end>

<start>
bottles<tab>1
terrier<tab>1
wildlife<tab>1
[...]
monarch<tab>2
terrier<tab>5
pitcher<tab>1
<end>

Neuron 489
Explanation of neuron 489: the main thing this neuron does is find natural habitats and various types of animals found in those habitats.
Activations:
<start>
carriage<tab>unk
regard<tab>unk
strawberry<tab>
[...]
blanca<tab>unk
dome<tab>unk
mask<tab>unk
<end>

where <tab> is the tab token, and "[...]" indicates that we truncated the prompt.

## D.6 Prompts for Weight-CLIP Based Explanation Generating

To keep consistency, we will color-code the below prompt with the same colors as the prompts above, namely: the system prompt in olive, the few-shot questions in blue, the few-shot replies in purple and the target neuron prompt in teal.

We're studying neurons in a computer vision neural network. Each neuron looks for some particular thing in an image. Look at the classes that the neuron is positively contributing towards and summarize in a short text what the neuron is looking for (the key thing most of classes have in common). Don't list examples of classes. Keep in mind that these are images, so the information connecting them might not be related to the object itself, but rather to what can be found in images of the object (i.e. fish & water).
The activation format is token<tab>weight. Activation values range from 0 to 10. A neuron finding what it's looking for is represented by a high positive value. The higher the value the stronger the match. You should not list objects, but rather try to find what they have in common.

Neuron 2
Activations:
<start>
mating<tab>7
mating<tab>7
mating<tab>7
organ<tab>7
parrot<tab>7
[...]
motif<tab>8
vendors<tab>8
watersports<tab>9
wheel<tab>9
font<tab>10
<end>

Explanation of Neuron 2 behavior: the main thing this neuron does is find large dogs and cats.

Neuron 1
Activations:
<start>
beverages<tab>7
crane<tab>7
dogs<tab>7
fluffy<tab>7
golf<tab>7
[...]
pitcher<tab>8
slashdot<tab>8
parrot<tab>9
vessels<tab>9
tractors<tab>10
<end>

Explanation of Neuron 1 behavior: the main thing this neuron does is find work vehicles, work dogs and rare wild birds.

Neuron 489
Activations:
<start>
bucket<tab>7
chestnut<tab>7
cobra<tab>7
dashboard<tab>7
fisheries<tab>7
[...]
towing<tab>8
wetland<tab>8
secure<tab>9
trout<tab>9
terrier<tab>10
<end>

Explanation of Neuron 489 behavior: the main thing this neuron does is find

where <tab> is the tab token, and "[...]" indicates that we truncated the prompt.

