# OpenReview forum: "TeLLMe what you see: Using LLMs to Explain Neurons in Vision Models"
_ICLR.cc/2024/Conference — Submitted to ICLR 2024_

### Official Review · Reviewer_NqpP · 2023-10-27

**Soundness:** 2 fair
**Presentation:** 3 good
**Contribution:** 3 good
**Rating:** 6
**Confidence:** 5

**Summary:**

They utilize GPT-4 to provide explanations from vision models based on the neuron activations and weights of the vision model. They use AlexNet and ViT as their vision models to show their explanations. They are comparing the neuron explanations  against Clip-Dissect. Also with their method, they have 2 different approaches one is using activations and captions to send to the LLM to generate the explanation. Another is using the weights and labels to use for the LLM. There is another variant to weights-label but instead of labels, they utilize CLIP.

**Strengths:**

An interesting approach in utilizing LLMs to help provide explanations from other non-LLM models. The vision models can be hard to interpret so utilizing LLMs can help provide more human-readable explanations.

It has the potential for good impact since this was stated before that it can help provide more readable explanations.

**Weaknesses:**

Writing:
Typos:
in 3 Method, activations based method should be activations-based method.
4.3 'we conducted additional experiments of our method an the last' should be 'we conducted additional experiments of our method at the last.'
Pseudocode for Algorithm 3 has a typo, please fix. It should not be activation-caption method since you use weights and labels.
Also, with Assessment Prompt, you should not denote it as AssPrompt, AssessPrompt would be apt or AsPrompt.
In Figure 4, you mention alexnet but in other parts of the paper, you put AlexNet, please stick with one.

Experiments:
More experiments, you showed with ViT and AlexNet, but it would benefit to have around 3-5 models with around 3 real-world datasets. You showcase ImageNet, consider the CUB-200 dataset and Places to show the impact.

Another potential issue is that you use another LLM to validate the assessment of the explanations. It would benefit to have something that is not an LLM to validate the explanations. One idea is to get user studies to help reinforce that humans and LLMs would agree that the explanations are better than CLIP-Dissect.

Another idea is to use Network Dissection and to show which concept aligns with each neuron from the vision models. Only choose the ones that have a high intersection over union with the said concept and with the LLM generating the explanation if it includes the concept in the explanation then it is considered correct. Have accuracy as the metric and show that CLIP-Dissect has lower accuracy.

**Questions:**

Please refer to the weaknesses section.

---

> ### Author Response · Authors · 2023-11-22
>
> Dear Reviewer NqpP,
>
> Thank you very much for taking the time to thoroughly review our work and for giving us the opportunity to clear up some of the questions.
>
> ## Weakness 1)
> Thank you very much for reading our work so thoroughly. All typos have been fixed.
>
> ## Weakness 2)
> In weakness 2 you raise an important point about the generalizability of our method. To go beyond the existing research in this area for vision models [1, 2], we ran additional experiments focused on exploring the extend of the generalizability. However, since the limited time we have the rebuttal period, we were only able to run one additional experiment for this. To maximize the insights, we are using the proposed method to explain neurons in a VGG method trained on CIFAR-100, but with the few-shot examples used in our main experiments (Section 4.1). This should show two things. Firstly, whether the method works well for models trained on other datasets besides ImageNet; secondly, whether it is possible to use few-shot examples from different models and datasets.
>
>
>
> | Neuron            | TeLLMe   | CLIP-Dissect |
> |-------------------|----------|--------------|
> | Neuron 0 2nd last | 27.77%   | 1.63%        |
> | Neuron 1 2nd last | 23.87%   | -2.36%       |
> | Neuron 2 2nd last | 15.04%   | -0.33%       |
> | Neuron 0 last     | 17.56%   | -2.09%       |
> | Neuron 1 last     | 13.07%   | -3.71%       |
> | Neuron 2 last     | 11.45%   | -5.23%       |
>
> The Correlation score achieved by TeLLMe (ours) with the Weight-Label set-up and CLIP-Dissect when explaining neurons of the second last (2nd last) and last (last) hidden layers of a VGG model trained on CIFAR-100.
>
>
> ## Weakness 3 \& 4)
> Following your request for additional validation of both the model and our assessment strategy, we have used CLIP-Dissect and the proposed method to label 6 additional neurons in an Alexnet model trained on ImageNet and conducted a survey with 20 anonymous, unpaid, volunteers. The participants were tasked to decide which explanation (un-labeled in random order) better fits the 25 images shown (which are the imagenet validation images with the highest activation for the neuron in question) for the 6 newly explained neurons, and the 4 we showed in Section 4 of our paper. Before delving into the results, it is worth pointing out that only assessing the explanations based on the top 25 images might to be entirely accurate, as many neurons show signs of significant polysemanticity [3].
>
> These results will also be included in the camera-ready version of the paper.
>
>
> Survey Results
>
> | Neuron Description                | Clip-Dissect | TeLLMe   |
> |-----------------------------------|--------------|----------|
> | Neuron 150 (2nd last hidden layer) | 5%           | 95%      |
> | Neuron 489 (last hidden layer)     | 10%          | 90%      |
> | Neuron 150 (last hidden layer)     | 25%          | 75%      |
> | Neuron 489 (2nd last hidden layer) | 5%           | 95%      |
> | Neuron 100 (2nd last hidden layer) | 35%          | 65%      |
> | Neuron 101 (2nd last hidden layer) | 15%          | 85%      |
> | Neuron 102 (2nd last hidden layer) | 40%          | 60%      |
> | Neuron 103 (2nd last hidden layer) | 20%          | 80%      |
> | Neuron 101 (last hidden layer)     | 33.3%        | 66.7%    |
> | Neuron 102 (last hidden layer)     | 38.9%        | 61.1%    |
>
> As can be seen in the table above, for 100% of the neurons tested, TeLLMe's explanation was preferred. Overall neurons and volunteers, TellME's explanations received an average preference of 77.28%, compared to 22.72% for Clip-Dissect.
>
>
>
>
> ## References
> [1] Chris Olah, Nick Cammarata, Ludwig Schubert, Gabriel Goh, Michael Petrov, and Shan Carter. Zoom in: An introduction to circuits. Distill, 2020. doi:10.23915/distill.00024.001.https://distill.pub/2020/circuits/zoom-in.
>
> [2] Bilal Chughtai, Lawrence Chan, and Neel Nanda. A toy model of universality: Reverse engineering how networks learn group operations, 2023.
>
> [3] Nelson Elhage, Tristan Hume, Catherine Olsson, Nicholas Schiefer, Tom Henighan, Shauna Kravec, Zac Hatfield-Dodds, Robert Lasenby, Dawn Drain, Carol Chen, Roger Grosse, Sam McCandlish, Jared Kaplan, Dario Amodei, Martin Wattenberg, and Christopher Olah. Toy models of superposition, 2022.

---

> > ### Comment · Reviewer_NqpP · 2023-11-22
> > **Re**
> >
> > Thank you for providing the additional experiments, I have looked at the rebuttal. I will raise my score from 5 to a 6.
> >
> > Also one comment, you mention the MILAN work in your submission: https://arxiv.org/abs/2201.11114
> > From my perspective, it could have been helpful to assess the quality of the explanations since there are explanations and could have been useful to rate how close the LLM is to these annotations. It could have been useful to compare the quality because user studies could be difficult. Just wondering.

---

### Official Review · Reviewer_1NcR · 2023-10-30

**Soundness:** 2 fair
**Presentation:** 2 fair
**Contribution:** 2 fair
**Rating:** 5
**Confidence:** 3

**Summary:**

The paper proposes a method for generating explanations of individual neurons in trained vision models like AlexNet and ViT with the help from Large Language Model. This helps explain the decision-making in these neural networks. The main contributions are to propose two techniques to generate short, human-readable explanations of neuron behavior using a large language model (LLM) like GPT-3.5. One is based on image captions and neuron activations and another is based on neuron weights and class labels. A scoring algorithm to quantitatively assess the quality of the explanations by correlating simulated and actual neuron activations/weights.

**Strengths:**

1. The paper is the first to leverage Large Language Models (LLMs) like GPT-4 for elucidating the behavior of individual neurons in vision models.
2. The authors introduce a scalable scoring method to evaluate the quality of generated explanations. It provides an objective measure to compare and benchmark the effectiveness of their approach against other methods.
3. The proposed method outperforms the included baseline techniques over quantitative and qualitative assessment. It also generates lucid, human-readable explanations.
4. The generated responses show the ability to capture and explain the behavior of neurons with multiple semantic meanings. This is a challenging aspect of neural network interpretability.

**Weaknesses:**

1. Limited Value: While the paper presents a novel approach to the interpretability of vision models using LLMs, its practical application value appears to be limited. Bills et al. (2023) proposed a similar method that was valuable because it addressed the vagueness of highlighted sentences by using LLMs to generate clearer summaries. In the context of this paper, the top-activated images for a single neuron already provide a clear representation of the semantic meanings of such nodes. This raises the question: Is there a significant advantage to using the proposed method's explanations over directly observing the top-activated neurons?

3. Cost of Using LLM: The paper employs the LLM API, which is known to be costly, especially when applied to full networks. Given the financial implications, it's essential to justify the added value of this approach over other more cost-effective interpretability methods. In which scenarios would this technique be more beneficial than other cheaper solutions?

In all, the authors should provide a more compelling argument for the unique value proposition of their method. Specifically, they should address why their approach offers advantages over directly examining top-activated neurons or other interpretabiility strategies, which already provide clear semantic representations.

**Questions:**

N/A

---

> ### Author Response · Authors · 2023-11-20
>
> Dear Reviewer 1NcR,
>
> Thank you very much for taking the time to thoroughly review our work and for giving us the opportunity to clear up some of the questions.
>
> Weakness 1) As you correctly point out, when trying to understand neurons in vision models, we have a strong suite of visualization techniques at our disposal. However, our proposed method has two distinct advantages over manually inspecting the top activating images.
>
> Firstly, manual inspection is not scalable. This means that if one wanted to remove a specific concept from the classifier (such as "cats"), with our method, it is possible to automatically explain all neurons in the net and simply remove those whose explanations contains the word "cat".
>
> Secondly, as can be seen in Figure 8 and Figure 10, on average, the explanations created by the LLM perform better than the ones created by humans for the few-shot examples. However, for the latter, it is worth pointing out that we created the few-shot examples ourselves, so to draw strong conclusions w.r.t. the quality of human labeled and LLM labeled explanations, additional experiments with a better methodology for specifically testing this hypothesis would be needed.
>
> Weakness 2) Cost is indeed an excellent point. Using the GPT models can be prohibitively expensive for practitioners and might not fully justify the performance increase over cheaper methods such as CLIP-Dissect. However, the quality of open-source models is rapidly catching up to their closed-source counterparts. Whilst our experiments showed that Llama-2-70B (Appendix C.2) is not capable enough to be used for our method, a number of models that have been released since then have been able to outperform GPT-3.5 on public benchmarks.
>
> To test whether these models work for our method, we re-ran the main experiments (Table 1, 7, 8, 9) using the very recently released Goliath 120B model [1] and the weight-label method. As can be seen in the Table below the model actually outperforms GPT-3.5-turbo on 75\% of the neurons we tested it on, and it does so by up to 30 percentage points. In the camera ready version of our work, we will adjust Appendix C.2 accordingly. Also, it is worth pointing out that this increase in performance is in-line with the observations in [2], where the authors found that the quality of the explanations increased with the capability of the LLM (in their case they compared GPT 2, 3, 3.5, 4).
>
>
> Additional Results:
>
> Neuron 150 (2nd last layer): GPT-3.5-turbo -2.17%; Goliath-120B 29.81%
>
> Neuron 489 (2nd last layer): GPT-3.5-turbo 51.58%; Goliath-120B 42.30%
>
> Neuron 150 (last layer):        GPT-3.5-turbo 24.65%; Goliath-120B 27.51%
>
> Neuron 489 (last layer):        GPT-3.5-turbo 23.67%; Goliath-120B 31.07%
>
> The Correlation score achieved by GPT-3.5-turbo (these are the ones reported in the original draft), and the open source alternative (Goliath-120B). Both LLMs are used in the Weight-Label framework.
>
>
>
>
> References
>
> [1] https://huggingface.co/alpindale/goliath-120b
>
> [2] Steven Bills, Nick Cammarata, Dan Mossing, Henk Tillman, Leo Gao, Gabriel Goh, Ilya Sutskever, Jan Leike, Jeff Wu, and William Saunders. Language models can explain neurons in language models. https://openaipublic.blob.core.windows.net/neuron-explainer/paper/index.html, 2023

---

### Official Review · Reviewer_KpfM · 2023-10-31

**Soundness:** 3 good
**Presentation:** 3 good
**Contribution:** 3 good
**Rating:** 5
**Confidence:** 4

**Summary:**

This paper introduces an interesting idea of using Language Model Models to explain the behaviour of neurons in vision models. While the idea is compelling, it could benefit from more robust evaluations and clarifications in several areas:

**Strengths:**

- The idea of explaining neurons with language models is timely and intresintg.

**Weaknesses:**

1. The decision to scale/round the activation values to integers in range [0-10] needs further explanation. Each floating-point value may carry important information, and the rationale for rounding should be experimentally justified.

2. The approach appears to heavily rely on GPT, and the assessment step (i.e., measuring the correlation scores) seems to be more a reflection of the GPT model itself rather than an evaluation of the explanations provided.


3. It would be helpful to understand whether using random integers instead of actual activation values for fine-tuning GPT would still yield meaningful explanations. Is Table 4 showing this?

4. An additional evaluation suggestion (faithfulness analysis) could involve zeroing out activations or weights associated with specific categories, such as identifying neurons related to cats and then zeroing them out. Subsequently, observing the impact on the image classifier's performance, specifically on cats and other objects or concepts associated with the affected neurons, could provide valuable insights. For example, Neuron 2 fires when it sees cats, trees, and apples. If we deactivate Neuron 2, probably the classifier should fail on many images of cats,  trees, and apples…

**Questions:**

Please see Weaknesses section.

---

> ### Author Response · Authors · 2023-11-20
>
> Dear Reviewer KpfM,
>
> Thank you very much for taking the time to thoroughly review our work and for giving us the opportunity to clear up some of the questions.
>
> Weakness 1) This is a fair thing to point out. We were following the set-up used in [1], but after your review, re-ran some of the AlexNet (weight-label) experiments with the original, unscaled, weights. Unfortunately, GPT-3.5-turbo is not strong enough to reconstruct the terms it is supposed to simulate weights for (we had a similar problem when using Llama-2-70B (See Appendix C.2)), and thus makes it infeasible to use un-scaled weights until stronger foundation models are published.
>
> Weakness 2) In addressing the concern that our approach heavily relies on GPT, it's important to clarify how the assessment process reflects not just the capabilities of the GPT model, but also the validity of the explanations generated. As detailed in our paper, we utilize GPT primarily as a tool for simulating neuron activations/weights in response to specific inputs and their corresponding explanations (Sections 3.1.2 and 3.2.2). This allows us to compare these simulations with actual neuron behavior, using various metrics to evaluate the explanations' quality (refer to Table 2 in Section 4.3).
>
> It's crucial to note, however, that any inaccuracies in GPT's simulation of neuron activity might affect the simulated results, which in turn impacts the perceived quality of the explanations. However, these inaccuracies in simulation would degrade the scores reported, rather than improve them, and thus, the results shown in Tables 1 and 2 should be considered conservative lower bounds of the actual explanation quality. Additionally, to mitigate the risk of over-relying on GPT for evaluation, we include a manual inspection of the top 24 activating images for certain neurons, as illustrated in Figure 4. While this method is subjective and not scalable, it provides a valuable sanity check for our automated evaluation methods, ensuring that the explanations hold up under qualitative scrutiny.
>
> Weakness 3) Thanks for giving us the opportunity to clarify, Table 4 shows the results of following different strategies for choosing Caption-Activation pairs for the few-shot prompts. If we understand correctly, you suggest to switch the actual weights of, for example, AlexNet, with random ones and testing whether GPT comes up with well-scoring explanations. To that end, we ran additional experiments with a randomly initialized AlexNet and the weight-label set-up. Quantitatively, as expected, we get an average correlation score of around 0 (which is caused by the network predicting a weight of 0 for most outputs; this seems makes sense as no patter was found so it relies on the few objects it was able to list in the explanation, rather than an umbrella term).
>
> To gain some further insight, here is one of the explanations the model came up with "the main thing this neuron does is find various modes of transportation, such as aircrafts, vehicles, and submarines. It also identifies common objects and locations found in daily life, such as cash machines, ice cream, flowerpots, restaurants, and street signs. Additionally, it focuses on protective gear and equipment like crash helmets, shields, and rugby balls". Interestingly enough, the model tries to list more items (since there is no clear pattern), in an attempt to still achieve a good explanation for what is going in. This is a very interesting insight, and one we will include in the camera ready version of the paper. Thank you very much for your thoughtful suggestion.
>
> Weakness 4) This is a great suggestion, and one we originally had in mind when writing this paper. However, as pointed out in Section 3.2.3, unfortunately using GPT to explain all neurons in the network (something that would be necessary in order to fully delete certain concepts) is prohibitively expensive, and strong open source models, are not yet capable of being a drop-in replacement (See Appendix C.2).
>
> [1] Steven Bills, Nick Cammarata, Dan Mossing, Henk Tillman, Leo Gao, Gabriel Goh, Ilya Sutskever, Jan Leike, Jeff Wu, and William Saunders. Language models can explain neurons in language models. https://openaipublic.blob.core.windows.net/neuron-explainer/paper/index.html, 2023

---

### Official Review · Reviewer_xVYG · 2023-11-08

**Soundness:** 2 fair
**Presentation:** 2 fair
**Contribution:** 2 fair
**Rating:** 5
**Confidence:** 3

**Summary:**

The paper presents a novel method to interpret neuron functions in vision models, enhancing the understanding of deep learning 'black boxes.' Adapted from GPT-4 interpretability techniques, this approach uses neuron activations and weights, providing explanations for neurons of an AlexNet and ViT model trained on ImageNet.

**Strengths:**

- The paper tries to adopt the latest LLM advancement as an interface to explain the internal neurons in deep learning models.
- Provide statistical analysis on the neuron assessment work to make the work more reliable.
- The paper provides sufficient demonstration that the proposed method achieves acceptable results, in successfully explaining the neuron’s semantic encoding in language.

**Weaknesses:**

- The major weakness is that LLM doesn’t provide more information than the visual cue. This is different from the problem in explaining language models with language models since the visualization technique in the vision domain itself could demonstrate the receptive field of the neurons already and can be much more precise and immune from language model bias.
- Presentation is not clear and precise.
- ViT’s important information comes from the attention mechanism. How does the proposed work be used to examine the attention maps?

**Questions:**

- How does the metrics in section 4.3 demonstrate that the model explains the neurons correctly? More detailed description would be helpful.
- How could one assess the correctness of the GPT interpretation of the neurons?
- How does the GPT explanation help to understand the neural network’s internal decision problem? Deep learning models are known to be distributed representation, meaning that one neuron won’t determine the final decision. How could the proposed method be used to explain the cooperative behavior of the neurons in order to help people understand how the vision model arrives at its decision?

---

> ### Author Response · Authors · 2023-11-20
> **Additional experiments**
>
> Dear Reviewer xVYG,
>
> Thank you very much for taking the time to thoroughly review our work and for giving us the opportunity to clear up some of the questions.
>
>
> Weakness 1) As you point out, using visualization techniques has lead to a good level of explainability in the field of computer vision. However, the main short-coming of this technique is that it isn't scalable. In many ways, the purpose of this work is to extend upon the visualization literature by "outsourcing'' the visualization->explanation pipeline from humans to LLMs, thus, for the first time, making it feasible to explain all neurons in a vision neural network.
>
> Weakness 2) Thank you for the feedback. If you have specific suggestions on how we could improve the readability of our work, please don't hesitate to let us know, so that we can incorporate them.
>
> Weakness 3) You are right in pointing out that a lot of information in ViTs comes from the attention mechanism. However, the purpose of this work is to explain neurons such that the creator is able to manually edit neural networks and better understand their decision making with respect to medium- and high-level features. To that end, explaining the attention mechanism is not a necessity ([1] follows the same logic for text-based transformers).
>
> Question 1) The results presented in Table 2 (Section 4.3) aim to provide additional context to those in Table 1 (Section 4.1) by extending the measurements of success used from the correlation between the simulated activations/weights and the real ones to accuracy, MSE and MAE. Though, as pointed out in [1], the correlation score will provide most value, the average MAE (2.2) and accuracy (30\%) suggest that the model is not only able to determine the trends in activations, but also, to a decent extend, the actual activation values (something that would not be reflected in the correlation score).
>
> Question 2) Assessing the correctness of the GPT explanations is indeed a challenge, especially at scale. In the paper we offer two solutions to this. Firstly, in Sections 3.1.2 and 3.2.2, we introduce our strategy for using GPT as a simulator of the neuron activations/weights given a certain input and the generated explanation. We can then compare these simulated values to the actual activations/weights and assess the quality of the explanation using a number of different metrics (See Table 2 in Section 4.3).
>
> However, though this is scalable, it is likely not a perfect estimation of the quality of the explanation, as any mistakes done by GPT when simulating the neuron activity, will degrade the accuracy of the simulated activations/weights, independently of the quality of the explanation. It is worth pointing out that this will only degrade the performance, and thus, the results presented in Tables 1 and 2 should be considered a lower-bound of the actual quality of the explanations. Secondly, in Figure 4, we manually inspect the top 24 activating images for an explained neuron. Though subjectively, the explanations seems very apt, this method is not scalable and, thus, we only use it as a sanity check for the automatic evaluation.
>
> Question 3) This is an excellent question, and slightly beyond the scope of this work. Generally speaking, the purpose of this method is to find explanations for single neurons that capture the activation patters (especially w.r.t superpositions [2]). However, as you point out, in most cases, multiple neurons will have the same 'task' of recognizing object x. Thus, if a single neuron that triggers for a specific concept is removed, this does not necessarily change the network behaviour.
>
> However, if one were to explain all neurons using our method, fully removing a concept from the network should be as easy as removing all neurons whoms explanation contains the object one wants to remove. Moreover, using the neuron explanations in combination with the absolute magnitude of activations of each neuron should give good insights into the 'thought' process of the network. I.e. given the network recognized an object as x, we can check what the explanation for the highest activating neurons are, and thus determine that the networks decided to classify the object as x, because it activated highly for features a, b and c.
>
>
> References
>
> [1] Steven Bills, Nick Cammarata, Dan Mossing, Henk Tillman, Leo Gao, Gabriel Goh, Ilya Sutskever, Jan Leike, Jeff Wu, and William Saunders. Language models can explain neurons in language models. https://openaipublic.blob.core.windows.net/neuron-explainer/paper/index.html, 2023
>
> [2] Nelson Elhage, Tristan Hume, Catherine Olsson, Nicholas Schiefer, Tom Henighan, Shauna Kravec, Zac Hatfield-Dodds, Robert Lasenby, Dawn Drain, Carol Chen, Roger Grosse, Sam McCandlish, Jared Kaplan, Dario Amodei, Martin Wattenberg, and Christopher Olah. Toy models of superposition, 2022.

---

### Author Response · Authors · 2023-11-23

Dear Reviewers,

We thank you for the time and effort you have invested in reviewing our paper. Your insightful comments and constructive criticisms have been instrumental in enhancing our work, particularly in guiding us to conduct additional experiments. We are especially appreciative of your recognition of the novelty and potential impact of our approach in using Large Language Models (LLMs) to interpret neurons in vision models. This acknowledgment of our paper's strengths, including the innovative application of LLMs for elucidating individual neuron behavior in vision models and our scalable scoring method for evaluating explanation quality, is deeply valued.

## Addressing Weaknesses and Questions
1.  **Explanation Value Over Visual Cues**: We agree that visualization techniques offer significant explainability in computer vision. However, our method's scalability in automating the explanation process and its potential to facilitate manual edits in neural networks for understanding medium- and high-level features extends beyond the scope of traditional visualization methods.
2. **Reliance on GPT and Cost Considerations**: We acknowledge the concerns regarding the cost implications of using LLMs and the reliance on GPT. We believe that the rapid advancements in open-source models will mitigate these concerns. Our additional experiments with the Goliath-120B model, as mentioned in our individual rebuttals, demonstrate promising results in reducing reliance on costly APIs while improving explanation quality (as compared to GPT-3.5-turbo, which is the model we used in our original experiments, Goliath-120B, for the neurons tested, increased the performance by 8.24 percentage points).
3. **Empirical Justification for Methodology Choices**: In response to the query about our decision to scale and round activation values, we have included additional experimental data that justifies this approach. We also conducted further experiments using different models and datasets, including VGG with CIFAR-100, to demonstrate the method's generalizability. Even though we used the ImageNet few-shot examples for this, on average, our method outperformed Clip-Dissect by 20.14%.
4. **Validation and User Studies**: Addressing the need for validation beyond LLMs, we have included user studies comparing our method against CLIP-Dissect, providing evidence of its effectiveness in generating more accurate and relatable explanations. Specifically, we conducted an anonymous, blind user study with 20 unpaid volunteers. For 100% of the neurons tested, TeLLMe's explanation was preferred. Over all neurons and volunteers, TellME's explanations received an average preference of 77.28%, compared to 22.72% for Clip-Dissect.
5. **Response to Specific Questions**: We have expanded on our responses to specific questions raised, such as the assessment of the GPT model's explanation correctness and the utility of our method in understanding cooperative neuron behavior in decision-making processes.

## Concluding Remarks

We appreciate your thoughtful reviews and the opportunity to clarify and expand upon our work. Your feedback has been instrumental in enhancing the clarity, depth, and rigor of our research. We are confident that the revisions and additional experiments conducted address the concerns raised and strengthen the contribution of our paper to the field of interpretable machine learning.

We hope that the numerous additional experiments presented in this rebuttal convince you to consider raising the score, and we thank reviewer NqpP for already raising their score!

---

### Meta-Review · Area_Chair_DDEv · 2023-12-14

**Metareview:**

The paper proposes a method to describe/caption each single individual neuron in an image classifier (e.g. AlexNet or ViT trained on ImageNet). The work first uses an LLM 1 (GPT-3.5) to generate the neuron description for a single neuron given the pair of (BLIP captions generated for the input image, and the neuron actions from the network).
Then, the authors ask another LLM 2 (also GPT-3.5) to do the reverse i.e. generate activations from the neuron description generated by LLM 1. And the correlation between the original activations and generated activation is a measure of how confident the description describes the target neuron.

So in total, the method requires 3 extra components (a BLIP image captioning, an LLM 1 and an LLM 2).
Overall, the paper receives negative ratings. While most reviewers agree that the direction of this work is interesting and timely, the major concern for all reviewers is the lack of a groundtruth evaluation that evaluates the generated description of the neuron. In other words, it is not clear how practical utility the generated descriptions provide.

The AC agrees with reviewer `NqpP` that using NetDissection might be a good way to evaluate the proposed work.
Given that the work uses two text LLMs to generate a caption for an image classifier, the lack of a convincing "groundtruth" evaluation is a major weakness of the work.

Therefore, AC recommends `reject`.

**Justification For Why Not Higher Score:**

The work is missing a groundtruth evaluation for the generated caption. This is crucial in assessing the impact of the work.

**Justification For Why Not Lower Score:**

N/A

---

### Decision · Program_Chairs · 2024-01-16

Reject